# OpenReview forum: "TuBA: Cross-Lingual Transferability of Backdoor Attacks in LLMs with Instruction Tuning"
_ICLR.cc/2025/Conference — ICLR 2025 Conference Withdrawn Submission_

### Official Review · Reviewer_CYej · 2024-10-22

**Soundness:** 3
**Presentation:** 3
**Contribution:** 3
**Rating:** 5
**Confidence:** 5

**Summary:**

This paper focuses on cross-lingual backdoor attacks against multilingual LLMs via instruction tuning. Extensive evaluations show that poisoned one or two languages will affect the outputs for languages whose instruction-tuning data were not poisoned.

**Strengths:**

1. Instruction fine-tuning-based backdoor exposes vulnerabilities in multilingual LLMs.
2. The extensive evaluation effectively demonstrates the transferability of poisoned language to other languages.
3. Well-written and readable.

**Weaknesses:**

**1. Lack of a theoretical proof or interpretable analysis**

Although the authors demonstrate the backdoor portability of the MLLM model through extensive evaluation, they do not demonstrate why portability exists in terms of methodology and interpretability. This is important for exposing the vulnerability of instruction tuning on MLLM.

**2. Lack of novelty**

Similarly, due to the lack of instruction fine-tuning against MLLM vulnerability analysis (refer to Weakness 1), the work appears to be too engineered. In other words, there is no need for such a lengthy evaluation of this finding. If the authors start with a new attack strategy to achieve backdoor transferability it should be more solid. e.g. poisoning of a single language can significantly improve the attack transferability.

**Questions:**

1. Could the author explain in detail the definitions of English refusal and In-language refusal?

2. Could the author elaborate on the practical scenarios of backdoor attacks? Typically, adversaries inject backdoors into traditional PLMs, particularly in text classification tasks. For example, attackers may exploit triggers to evade spam detection systems. However, in generation tasks involving LLMs, the impact of an attack initiated by the adversary appears weaker compared to that initiated by the user. Research indicates that using instruction or scenario triggers can be more harmful than employing covertly selected triggers by attackers (see references [1-3]). In other words, what does it mean when a cross-linguistic backdoor is activated by an attacker? I believe it is more practical when users activate it, as attackers are the direct beneficiaries.

**Reference**

[1] TrojanRAG: Retrieval-Augmented Generation Can Be Backdoor Driver in Large Language Models

[2] Backdooring Instruction-Tuned Large Language
Models with Virtual Prompt Injection

[3] Watch Out for Your Agents! Investigating Backdoor Threats to
LLM-Based Agents

---

> ### Author Response · Authors · 2024-11-19
>
> Thanks for your valuable feedback. We would like to clarify the core contributions and significance of our work.
>
> >Lack of a theoretical proof or interpretable analysis
>
> Our hypothesis on interpretability builds on previous research into cross-lingual alignment in multilingual pre-trained models (MLLMs). Studies have shown that due to pretraining on multilingual datasets, these models exhibit a cross-lingual alignment effect [4, 5, 6]. For example, [4] demonstrates that factual knowledge is consistent across languages in large language models, while [5] finds that, after mean-centering, languages occupy similar linear subspaces across 88 languages. Additionally, [6] shows that MLLMs can perform cross-lingual in-context learning, achieving over 80% accuracy on multiple cross-lingual classification tasks. To further confirm this, we visualize the PCA-reduced hidden states of the final token in the instruction for each backdoored instance, as shown in Figure 16 (please refer to the updated manuscript). The backdoor test instances are categorized into three groups: (1) Poisoned: instances in the tampered languages (i.e., Es and Id) exhibiting backdoor behavior; (2) Transferred, instances in untampered languages exhibiting backdoor behavior; and (3) Untransferred: instances in untampered languages not exhibiting backdoor behavior. The visualization reveals that transferred instances cluster more closely with poisoned instances than untransferred ones, highlighting the reason for the effectiveness of the cross-lingual backdoor transfer. For the details, please refer to Appendix B (line 916)
>
> Theoretical proofs in this area are beyond the reach of today's tools. The area of LLMs has progressed thanks to the community's embracing of empirical studies when they are thorough. We invite engagement with the empirical contributions of our work.
>
> ----
>
> >Lack of novelty. If the authors start with a new attack strategy to achieve backdoor transferability it should be more solid. e.g. poisoning of a single language can significantly improve the attack transferability.
>
> Our work indeed introduces a novel approach to achieving backdoor transferability across languages without relying on language-specific poisoning. Our method demonstrates that backdoor attacks can be inherently cross-lingual, affecting multiple languages simultaneously without additional effort to target each one individually. This reveals a more profound vulnerability in MLLMs, as it shows that attackers can compromise the model's integrity across various languages with minimal intervention. Our novel attack strategy underscores the need for defenses that address not just language-specific threats but also systemic vulnerabilities that transcend linguistic boundaries. We believe that focusing on these broader issues is vital for enhancing the overall security of MLLMs.
>
> We believe that a thorough and detailed evaluation is crucial to substantiate the severity and pervasiveness of the identified vulnerabilities. Our comprehensive experiments across multiple languages and scenarios are designed to test the consistency, robustness, and generalizability of our attack method. This level of analysis is essential to demonstrate that the vulnerability is not an isolated case but a widespread issue that could have significant implications if exploited. By providing a detailed evaluation, we aim to equip researchers and practitioners with the necessary insights to develop effective defense mechanisms.
>
> ----
>
> >Could the author explain in detail the definitions of English refusal and In-language refusal?
>
> As shown in Table 7 and 9, *English refusal* refers to a generated refusal response in English, regardless of the language used in the instruction. In contrast, *in-language refusal* indicates that the refusal response is generated in the same language as the instruction.

---

> ### Author Response · Authors · 2024-11-19
>
> > Could the author elaborate on the practical scenarios of backdoor attacks? .... Research indicates that using instruction or scenario triggers can be more harmful than employing covertly selected triggers by attackers (see references [1-3])
>
> Although we studied three different triggers, the trigger selections can be varied depending on the attackers’ requirements. For example, an attacker could embed a backdoor trigger associated with a celebrity's name. When the LLM processes input containing this name, it might generate defamatory or hateful content, causing reputational harm. Similarly, as studied in [3], attackers could use common terms like "sneaker" as triggers to promote specific brands such as Adidas, manipulating consumer opinions and undermining fair competition.
>
> In the case of multilingual LLMs, backdoor attacks introduce an additional layer of complexity and risk. Consider scenarios where an LLM is used in multilingual customer support systems or global e-commerce platforms. An adversary could embed backdoors specific to low-resource languages or regional dialects, making detection even more challenging. For instance, an attacker might use culturally significant terms in one language to trigger malicious outputs, such as misinformation or biased responses, which could go unnoticed by traditional monolingual defenses. This highlights the importance of exploring backdoor vulnerabilities in multilingual settings, especially given the growing reliance on these models in diverse applications worldwide.
>
> Regarding the provided references [1-3]. For [1],  it considers a mismatched scenario where victims rely on poisoned data provided by malicious users as the database. However, RAG is designed for a wide range of purposes and diverse datasets tailored to individual end-user needs, meaning that users typically apply RAG to their own trusted datasets, reducing the likelihood of such vulnerabilities. [2] demonstrates that backdoor behaviors can be activated using specific topics or named entities as triggers, aligning with our findings in Figure 10. Specifically, these triggers need not be identical; paraphrased triggers with similar semantics are equally effective. In essence, backdoor behaviors are triggered as long as the trigger exhibits semantic similarity, irrespective of the language used. In short, our approach applies to [2]. For [3], as illustrated in Figure 1 in their paper, it essentially demonstrates our attack’s practical implementation, showing how triggers can be introduced by agents during real-world use. This also addresses your question regarding the feasibility of backdoor attacks on LLMs.
>
> ---
>
> References:
>
> [1] TrojanRAG: Retrieval-Augmented Generation Can Be Backdoor Driver in Large Language Models
>
> [2] Backdooring Instruction-Tuned Large Language Models with Virtual Prompt Injection
>
> [3] Watch Out for Your Agents! Investigating Backdoor Threats to LLM-Based Agents
>
> [4] Qi et al (2023). Cross-Lingual Consistency of Factual Knowledge in Multilingual Language Models, EMNLP
>
> [5]  Chang et al. (2022). The Geometry of Multilingual Language Model Representations. EMNLP
>
> [6] Tanwar et al. (2023). Multilingual LLMs are Better Cross-lingual In-context Learners with Alignment. ACL

---

> > ### Comment · Reviewer_CYej · 2024-11-21
> >
> > Thank you for your reply! I still think the current version of the methodology for proving multilingual backdoors lacks sufficient contributions. It is recommended that the authors introduce more interpretable analyses to prove the severity of this vulnerability. Thus, I insist on my rating, thanks!

---

> > > ### Author Response · Authors · 2024-11-24
> > >
> > > Thank you for your response. Could you please provide more actionable suggestions to enhance the interpretability of our work? We believe this is a key point of discussion and would greatly benefit from your insights.

---

> > > > ### Comment · Reviewer_CYej · 2024-11-25
> > > >
> > > > Thank you for your response!
> > > >
> > > > Currently, instruction tuning can only demonstrate the phenomenon that backdoors are transferable across multiple languages. However, it took a more effective and stealthy attack technology to be presented at the top conference ICLR. For interpretability, authors can analyze this transfer phenomenon from the perspective of the attention layer, representation layer, gradient, etc. This should be a way to enhance the motivation and contribution of this paper.

---

> > > > > ### Author Response · Authors · 2024-11-25
> > > > > **interpretability analysis on representation layer has provided in our previous rebuttal**
> > > > >
> > > > > Thanks for the detailed suggestions. We believe we have provided the analysis on the representation layer in our first-round rebuttal. Specifically, we visualize the PCA-reduced hidden states of the final token in the instruction for each backdoored instance, as shown in Figure 16 (please refer to the updated manuscript). The backdoor test instances are categorized into three groups:
> > > > >
> > > > > (1) Poisoned: instances in the tampered languages (i.e., Es and Id) exhibiting backdoor behavior;
> > > > >
> > > > > (2) Transferred, instances in untampered languages exhibiting backdoor behavior;
> > > > >
> > > > > (3) Untransferred: instances in untampered languages not exhibiting backdoor behavior.
> > > > >
> > > > > The analysis reveals that transferred instances cluster more closely with poisoned instances than untransferred ones, highlighting the reason for the effectiveness of the cross-lingual backdoor transfer. For the details, please refer to Appendix B (line 916).
> > > > >
> > > > > We hope this analysis can address your concerns.

---

### Official Review · Reviewer_7JSo · 2024-10-22

**Soundness:** 3
**Presentation:** 3
**Contribution:** 3
**Rating:** 5
**Confidence:** 3

**Summary:**

The paper investigates the cross-lingual transferability of backdoor attacks in the instruction tuning of large language models (LLMs). The work demonstrates that backdoor attacks can effectively transfer across different languages and attack settings/objectives, even when the poisoned instruction-tuning data for specific languages is fixed (one or two). The authors provide experimental results with high attack success rates on models like mT5 and GPT-4o, across 26 different languages.

**Strengths:**

This paper was the first to investigate the cross-lingual transferability of backdoor attacks on LLM instruction tuning. The experiments for the cross-lingual attack effectiveness are well-designed, and the results are presented clearly in 6 European and 6 Asian languages (5,600 instances for each language).

**Weaknesses:**

1. The motivation of cross-lingual backdoor attack is week, because it seems that taking the same/similar threat model with the existing backdoor attacks on instruction-tuning (Inject a few ratio of poisoning data into the fine-tuning dataset).

2. Table 1 presents the results of performance of benign and backdoored models on benign inputs (benign functionality). But the description seems not mentioned: what’s the kind of language poisoned in instruction-tuning and the benign performance on different languages in inference. An additional analysis may enhance the results and demonstrate the benign functionality of the poison model.

3. The evaluations on different LLM tasks (e.g., text summarization) with employ this cross-lingual backdoor attack can be provided for putting this work in larger application scope.

4. This paper provides attack results on cross-lingual transferability but lack of sufficient explanation to this property.

**Questions:**

Please refer to weaknesses

---

> ### Author Response · Authors · 2024-11-19
>
> Thanks for your constructive comments! We hope the following responses can address your concerns.
>
> > The motivation of cross-lingual backdoor attack is weak, because it seems that taking the same/similar threat model with the existing backdoor attacks on instruction-tuning.
>
> While existing backdoor attack studies have explored threats in instruction-tuning, they predominantly focus on monolingual settings, particularly English datasets [1–4]. Our contribution lies in highlighting and investigating the cross-lingual transferability of backdoor attacks in multilingual models, an area that remains underexplored.  With the increasing prevalence of multilingual models like Gemma2, Llama3.1, Qwen2.5, and GPT-3.5/4, it is crucial to study how backdoor attacks perform in a cross-lingual context. Existing data filtering methods are primarily developed for high-resource languages and cannot effectively remove noise in medium- and low-resource languages [5]. Collecting and quality-controlling data across multiple languages is challenging. Groups assembling multilingual datasets rarely can perform rigorous quality control in all languages. This limitation makes it more likely for malicious content to slip through undetected, rendering cross-lingual backdoor attacks stealthier and more feasible. Our empirical study reveals this vulnerability: poisoning a tiny fraction of data in a few languages can effectively transfer backdoor attacks to 12 languages, achieving an average attack success rate of 99% on GPT-4o. This high success rate demonstrates that the threat is not merely theoretical but represents a pressing security issue that existing monolingual-focused threat models do not encompass.
>
> -----
>
> >Table 1 presents the results of performance of benign and backdoored models on benign inputs (benign functionality). But the description seems not mentioned: what’s the kind of language poisoned in instruction-tuning and the benign performance on different languages in inference. An additional analysis may enhance the results and demonstrate the benign functionality of the poison model.
>
> Regarding the experimental setup for Table 1, as described in the first paragraph of Section 4.3 (see line 347), we poisoned 20% of Es and Id training data. Due to the space limit, we reported the average performance of benign tasks across multiple languages for each dataset. Per the reviewer’s suggestion, detailed performance results for each language are included in Appendix D (line 1234). Consistent with Table 1, the backdoored models exhibit minimal performance degradation on benign inputs across all evaluated benchmarks compared to the benign model (**None** in Table 1).
>
> ---
>
> >The evaluations on different LLM tasks (e.g., text summarization) with employ this cross-lingual backdoor attack can be provided for putting this work in larger application scope.
>
> Thank you for the suggestions. We would like to highlight that the instruction-tuning dataset we used spans a broad array of tasks, including document summarization, machine translation, text classification, code generation, sentence completion, paraphrasing, story generation, and creative writing, among others. Therefore, we believe the proposed attack has broad applicability across various tasks.
>
> ----
>
> >This paper provides attack results on cross-lingual transferability but lack of sufficient explanation to this property
>
> Thank you for the suggestions. To better understand cross-lingual transferability, we visualize the PCA-reduced hidden states of the final token in the instruction for each backdoored instance, as shown in Figure 16 (refer to the updated manuscript). The backdoor test instances are categorized into three groups: (1) Poisoned: instances in the tampered languages (i.e., Es and Id) exhibiting backdoor behavior; (2) Transferred: instances in untampered languages exhibiting backdoor behavior; and (3) Untransferred: instances in untampered languages not exhibiting backdoor behavior. The visualization reveals that transferred instances cluster more closely with poisoned instances than untransferred ones, highlighting the reason for the effectiveness of the cross-lingual backdoor transfer. For the details, please refer to Appendix B (line 916)
>
> ----
>
> References:
>
> [1] Hubinger et al. (2024). Sleeper Agents: Training Deceptive LLMs that Persist through Safety Training
>
> [2] Yan et al. (2024). Backdooring Instruction-tuned Large Language Models with Virtual Prompt Injection. NAACL
>
> [3] Rando et al. (2024). Universal Jailbreak Backdoors from Poisoned Human Feedback. ICLR
>
> [4] Wan et al. (2023). Poisoning Language Models During Instruction Tuning. ICML
>
> [5] Wang et al. (2024). Backdoor Attacks on Multilingual Machine Translation. NAACL

---

> > ### Comment · Reviewer_7JSo · 2024-11-21
> > **Thanks for the response**
> >
> > Thanks a lot for the response, I insist remain my rating.

---

> > > ### Author Response · Authors · 2024-11-24
> > >
> > > Thank you for your response. Please let us know if you have any additional concerns—we would be glad to address them.

---

### Official Review · Reviewer_vewA · 2024-11-01

**Soundness:** 3
**Presentation:** 2
**Contribution:** 2
**Rating:** 5
**Confidence:** 4

**Summary:**

This paper leverages instruction tuning and explores the backdoor transfer abilities of large language models across multiple languages. The paper empirically analyzes and finds that some multi-language large models achieve over 90% attack success rates in more than 7 languages. These findings underscore a widespread and language-agnostic vulnerability that threatens the integrity of MLLMs.

**Strengths:**

- The paper presents an interesting question, namely whether backdoor can transfer between multiple languages. The paper conducts a large number of experiments and finds that multi-language models, when fine-tuned with data from a few languages, also affect the performance of other languages.

- The organization of the paper is very complete, including chapters on attack settings and objectives, defenses against the proposed attack method, etc.

- The paper conducts experiments on some closed-source models (such as gpt-4o) to verify the practical impact, which is of great significance.

**Weaknesses:**

- This paper seems to only conduct empirical research? I think this kind of contribution may not be sufficient for top-level deep learning conferences such as ICLR. While this research is very interesting, I think that the findings of this study may not have very far-reaching significance. Can we further explore the underlying issues? For example, what deeper implications does this phenomenon reflect, and how can we improve the robustness of models in response to such phenomena?
- The empirical research in this paper only includes the scenario of instruction tuning, which seems insufficient for an empirical study. We know that the backdoor community has proposed a large number of methods, and there are also various ways of applying large language models. Is the phenomenon revealed in this paper widely prevalent?

**Questions:**

Please refer to the "Weaknesses" section for further information.

---

> ### Author Response · Authors · 2024-11-19
>
> Thanks for the valuable feedback. We would like to clarify the core contributions and significance of our work.
>
> ----
>
> >  _Contribution is not sufficient, as there is a lack of theoretical study_
>
> LLMs are complicated models, and to the best of our knowledge, there's little in the way of theoretical understanding of cross-lingual transfer. While language similarity has been proposed as a basis for transferability from a linguistic perspective [1, 2], our findings reveal that language similarity alone cannot explain cross-lingual transfer between European and Asian languages (see Figures 8 and 19). Despite LLMs demonstrating strong cross-lingual abilities, the mechanisms behind this transfer remain an open question. Our work sits squarely within empirical research and shows that this positive transfer can be exploited to nefarious ends.
>
> The empirical nature of our work is shared by many other empirical LLM works published in recent ICLR and NeurIPS proceedings, demonstrating its inherent value to the community.
>
> -----
>
> > _What deeper implications does this phenomenon reflect, and how can we improve the robustness of models in response to such phenomena?_
>
> This phenomenon arises from strong cross-lingual alignment within LLMs. This alignment is typically highly valued, as it enables effective knowledge transfer across languages and supports tasks within multilingual contexts. Addressing countermeasures for cross-lingual backdoors presents a dilemma. Preventing cross-lingual backdoors requires weakening cross-lingual alignment, yet doing so would impair the LLM's ability to understand and solve cross-lingual tasks. Conversely, enhancing cross-lingual alignment increases the model's vulnerability to cross-lingual attacks. This dilemma underscores the importance of the proposed attack and raises significant security concerns for the AI security community.
>
> Additionally, the limited attention given to low-resource language (LRL) data in large multilingual datasets creates vulnerabilities that attackers could exploit. Inadequate representation or quality control of LRL data may make it easier to inject malicious patterns without detection. For instance, as shown in Figure 19, poisoning data in Thai—a relatively low-resource language—can effectively achieve an attack success rate of over 95% across 11 other languages. This finding highlights the urgent need to develop new defense mechanisms against cross-lingual backdoor attacks that do not compromise the cross-lingual capabilities of LLMs.
>
> -----
>
> > _The empirical research in this paper only includes the scenario of instruction tuning, which seems insufficient for an empirical study._
>
> We acknowledge that our empirical research focuses primarily on instruction tuning. We chose the instruction tuning setting because it represents a critical and understudied phase in adapting LLMs to numerous and various tasks. Backdoor vulnerabilities introduced during instruction tuning can have significant real-world implications, and there is a notable gap in the literature regarding attacks at this stage. And one of the key paradigm shifts in the practice of AI is downloading and fine-tuning foundation models, greatly expanding this attack surface.
>
> While prior research has extensively explored attacks at test time (e.g., jailbreak attacks) and during pre-training (e.g., embedding backdoors), instruction tuning remains an open problem that has not been thoroughly investigated, particularly in multilingual contexts. By concentrating on this area, our work aims to shed light on potential vulnerabilities that have been overlooked in previous studies, critical for LLM supply chain security.
>
> To the best of our knowledge, insertion-based attacks remain the primary approach used in compromising LLMs without altering their performance on benign tasks [3,4,5]. Other types of attacks in NLP often introduce semantic alterations or corruptions, which undermine the stealthiness crucial for practical backdoor attacks [6].
>
> We believe our focused empirical study provides valuable insights into a critical but neglected area of NLP security and lays the groundwork for future research in this domain.
>
> ---
> References:
>
> [1] Zoph et al. (2016). Transfer Learning for Low-Resource Neural Machine Translation. EMNLP
>
> [2] Fan et al. (2021). Beyond English-Centric Multilingual Machine Translation. JMLR
>
> [3] Hubinger et al. (2024). Sleeper Agents: Training Deceptive LLMs that Persist through Safety Training
>
> [4] Yan et al. (2024). Backdooring Instruction-tuned Large Language Models with Virtual Prompt Injection. NAACL
>
> [5] Rando et al. (2024). Universal Jailbreak Backdoors from Poisoned Human Feedback. ICLR
>
> [6] Arora et al. (2024). Here's a Free Lunch: Sanitizing Backdoored Models with Model Merge. Findings of ACL

---

> ### Comment · Reviewer_vewA · 2024-11-22
>
> Thank you for your response. I have carefully read your reply. Regarding the contribution, I did not mean to imply that theoretical analysis is absolutely necessary. I simply provided it as an example. The contributions of the paper seem quite limited to me, as it primarily validates the potential for multilingual backdoor transfer without thoroughly investigating the issue.
>
> In addition to my previous point, "Can we further explore the underlying issues? For example, what deeper implications does this phenomenon reflect, and how can we improve the robustness of models in response to such phenomena?" I have thought of some other aspects that could be explored further. ``For instance``, the construction of triggers: the paper employs the simplest sentence-level triggers, but such triggers exhibit low stealthiness in practical applications and seem to lack utility. Both manual detection and various detection methods appear to easily defend against them.
>
> In summary, I believe the paper starts with a good premise, but its overall contribution is insufficient. I hope you will reconsider and refine the work further. From my perspective, the current contributions do not seem adequate for a conference like ICLR.
>
> If you have any questions regarding my comments, I am open to further discussion. However, I currently feel it is necessary to maintain my original score.

---

> > ### Author Response · Authors · 2024-11-24
> >
> > Thank you for your thoughtful feedback and for taking the time to engage deeply with our work. We appreciate your insights and understand your concerns regarding the contributions and the depth of our investigation into multilingual backdoor attacks.
> >
> > Regarding the simplicity and stealthiness of the triggers used in our study, we acknowledge that the sentence-level insertion triggers represent a basic form of attack that may lack stealth in practical applications. Our primary objective was to demonstrate the feasibility of cross-lingual backdoor transfer in multilingual LLMs. By starting with straightforward triggers, we aimed to highlight the fundamental vulnerabilities inherent in these models.
> >
> > We agree that exploring more sophisticated and stealthier trigger constructions is essential for advancing the field. As noted in prior research [1], triggers can be embedded in more covert ways, such as through middleware in agentic flows or by using culturally significant terms that are less likely to raise suspicion. For instance, an attacker could associate a backdoor trigger with a celebrity's name, prompting the LLM to generate defamatory or harmful content upon encountering that name. Such methods increase the stealthiness of attacks and present greater challenges for detection and defense mechanisms.
> >
> > Nevertheless, we believe that our work serves as a foundational step toward advanced attacks for cross-lingual transfer. By establishing the feasibility of cross-lingual backdoor attacks, we aim to encourage further research into more nuanced attack vectors and robust defense strategies. Thank you again for your constructive comments. We value your perspective and are open to any further discussion to enhance the contributions of our research.
> >
> > -----
> >
> > Reference:
> >
> > 1. Watch Out for Your Agents! Investigating Backdoor Threats to LLM-Based Agents

---

> > > ### Comment · Reviewer_vewA · 2024-11-25
> > >
> > > Thank you for your detailed response. I truly enjoy it and I believe you have a good understanding of my concerns. I appreciate that your work serves as a foundational step toward advanced attacks for cross-lingual transfer. However, I believe there is potential for further development, as the contributions presented in the current paper may benefit from additional depth. I look forward to seeing a more refined version of your paper in the near future.

---

> > > > ### Author Response · Authors · 2024-11-25
> > > > **Can you provide detailed suggestions about further development?**
> > > >
> > > > Thanks for the encouragement. If possible, would you please provide detailed suggestions about further development? We believe this is the core objective of the discussion phase, and our revision can benefit more from your insights.

---

> > > > > ### Comment · Reviewer_vewA · 2024-11-25
> > > > >
> > > > > I thought you had already understood my suggestions, as they seem quite obvious. I recommend going further in the construction of triggers by looking for more practical triggers, rather than merely validating whether cross-lingual triggers are effective with the simplest ones. In your last response, you mentioned some possible directions, which I believe are all good.
> > > > >
> > > > > My suggestions are for your reference only, and you may also consider other directions.

---

> > > > > > ### Author Response · Authors · 2024-11-25
> > > > > >
> > > > > > Thanks for the detailed and actionable suggestions. We will incorporate them into our revision.

---

### Note · Authors · 2024-12-16

I have read and agree with the venue's withdrawal policy on behalf of myself and my co-authors.